# Impact of Two Resuscitation Sequences on Alveolar Ventilation during the First Minute of Simulated Pediatric Cardiac Arrest: Randomized Cross-Over Trial

**DOI:** 10.3390/healthcare10122451

**Published:** 2022-12-05

**Authors:** Laurent Suppan, Laurent Jampen, Johan N. Siebert, Samuel Zünd, Loric Stuby, Florian Ozainne

**Affiliations:** 1Division of Emergency Medicine, Department of Anesthesiology, Clinical Pharmacology, Intensive Care and Emergency Medicine, Faculty of Medicine, University of Geneva, Geneva University Hospitals, 1211 Geneva, Switzerland; 2ESAMB-École Supérieure de Soins Ambulanciers, College of Higher Education in Ambulance Care, 1231 Conches, Switzerland; laurent.jampen@edu.ge.ch (L.J.); florian.ozainne@edu.ge.ch (F.O.); 3Department of Paediatric Emergency Medicine, Geneva Children’s Hospital, Geneva University Hospitals, 1211 Geneva, Switzerland; johan.siebert@hcuge.ch; 4Service de la Protection et de la Sécurité, 2000 Neuchâtel, Switzerland; samuel.zund@ne.ch; 5Genève TEAM Ambulances, 1201 Geneva, Switzerland; l.stuby@gt-ambulances.ch

**Keywords:** pediatric cardiac arrest, bag-valve-mask ventilation, cardiopulmonary resuscitation, simulation study, study protocol, randomized trial, paramedics, chest compression fraction, alveolar ventilation

## Abstract

The International Liaison Committee on Resuscitation regularly publishes a Consensus on Science with Treatment Recommendations, but guidelines can nevertheless differ when knowledge gaps persist. In case of pediatric cardiac arrest, the American Heart Association recommends following the adult resuscitation sequence, i.e., starting with chest compressions. Conversely, the European Resuscitation Council advocates the delivery of five initial rescue breaths before starting chest compressions. This was a superiority, randomized cross-over trial designed to determine the impact of these two resuscitation sequences on alveolar ventilation in a pediatric model of cardiac arrest. The primary outcome was alveolar ventilation during the first minute of resuscitation maneuvers according to the guidelines used. A total of 56 resuscitation sequences were recorded (four sequences per team of two participants). The ERC approach enabled higher alveolar ventilation volumes (370 mL [203–472] versus 276 mL [140–360], *p* < 0.001) at the cost of lower chest compression fractions (57% [54;64] vs. 66% [59;68], *p* < 0.001). Although statistically significant, the differences found in this simulation study may not be clinically relevant. Therefore, and because of the importance of overcoming barriers to resuscitation, advocating a pediatric-specific resuscitation algorithm may not be an appropriate strategy.

## 1. Introduction

The incidence of out-of-hospital cardiac arrest (OHCA) is much lower in the pediatric population than among adults [1]. Their etiology is also different, since most pediatric OHCAs are the consequence of respiratory arrest [2]. Even though American [3] and European [4] resuscitation guidelines are both based on the same “Consensus on Science with Treatment Recommendations” (CoSTR) issued by the International Liaison Committee on Resuscitation (ILCOR) [5,6,7,8], these organizations recommend two fundamentally different strategies regarding the initial management of pediatric OHCA. Indeed, the American guidelines state that the management of pediatric OHCA should follow the same sequence as in adults and that rescuers should start by providing chest compressions immediately after diagnosing cardiac arrest [3]. Conversely, the European guidelines recommend the delivery of five rescue breaths before initiating chest compressions [4]. This difference spawns from a lack of definitive evidence, which prevented the ILCOR from issuing an unequivocal recommendation.

Because pediatric OHCA is generally the consequence of impaired oxygenation, rapidly reversing hypoxia should be the primary objective in such situations. Since oxygen uptake and delivery depend on the partial pressure of oxygen in the alveoli [9], rapidly optimizing alveolar oxygenation could result in positive outcomes. Alveolar ventilation is the main determinant of CO_2_ clearance [10], which is necessary to improve alveolar oxygen partial pressure [11]. This partial pressure is finally responsible for adequate tissue oxygenation, including brain oxygenation. In addition, improving CO_2_ clearance should help correct acidosis and its many harmful effects [12].

The impact of alveolar ventilation on newborns has previously been described [13,14], but there is little data regarding the impact of resuscitation sequences on alveolar ventilation in somewhat older infants. Gathering such data in a clinical setting is challenging, since designing a prospective, randomized clinical trial on a population of infants and children in OHCA is fraught with difficulties [15]. Therefore, simulation studies could help obtain relevant data even though such trials suffer from unavoidable limitations [16].

The hypothesis underlying this study was that the ERC resuscitation sequence should enable higher alveolar ventilation during the first minute of resuscitation in comparison with the AHA sequence. Therefore, its objective was to determine the difference in alveolar ventilation during the first minute of resuscitation according to the sequence used (AHA vs. ERC) in a pediatric model of OHCA.

## 2. Materials and Methods

This was a randomized, cross-over, superiority trial [17], designed using the SPIRIT (Standard Protocol Items: Recommendations for Interventional Trials) checklist [18] and reported according to the CONSORT guidelines [19]. It was registered on clinicaltrials.gov (NCT05474170).

### 2.1. Study Design and Sequence

The study sequence is detailed in the following subsections and in Figure 1. This study took place on the first Prehospital Research Day of the French part of Switzerland [20], which was held on 1 September 2022 in Neuchâtel, Switzerland.

#### 2.1.1. Participant Recruitment and Consent

Participant recruitment was conducted online. Emergency medical technicians (EMTs), paramedics, nurses, and physicians were all eligible for inclusion.

A web-based platform based on the Joomla 4 (Open Source Matters, New York, NY, USA) content management system was specifically created for the purpose of this study. The Event Booking 4 component (Joomdonation, Hanoi, Vietnam) was used to create 20 min time slots. Demographic data were collected during the registration process. Consent was gathered electronically. 

#### 2.1.2. Randomization and Concealment of Allocation

Since the objective of this study was to assess the impact of basic airway management and ventilation maneuvers only, there was no stratification, since all the professionals eligible for inclusion were considered to be equally proficient in basic airway management. Furthermore, all participants were given the opportunity to practice this skill on a manikin identical to the one used to perform the study prior to participating. This training was not time-limited, and participants were reminded of this training opportunity at least 15 min before entering the study room. Randomization was carried out using sealedenvelope.com. 

#### 2.1.3. Manikin and Equipment

A SimBaby manikin (Laerdal SimBaby, Laerdal Medical, Stavanger, Norway) was used. The SimBaby is a realistic manikin representing a 9-month-old infant. The manikin weighs 4.9 kg and is 71 cm tall. It is accompanied with a dedicated multiparameter monitor/defibrillator. Back compensation, using a folded blanket, was applied, and an appropriately sized bag-valve-mask (BVM) device was ready for use next to the manikin. The defibrillation pads were already attached. 

#### 2.1.4. Resuscitation Scenario

The participants were told that they were facing a 9-month-old infant who had suddenly collapsed. They were told that there was no foreign body airway obstruction and were informed that the infant was in cardiac arrest.

Each team of two performed a total of four resuscitation sequences, each lasting one minute. The first member of the team acted as leader for two successive resuscitation sequences. The resuscitation method which the leader was to follow (AHA or ERC) was determined according to the random allocation procedure described above. The scenario was identical for all resuscitation sequences. After completing these two sequences, team members exchanged their roles, and the new leader followed the same procedure.

The timer started (T0) when the manikin recorded the first resuscitation maneuver (chest compression or ventilation) and stopped after 60 s exactly.

#### 2.1.5. Blinding

Neither the participants nor the on-site investigators could be blinded as to the design of the study or even as to the sequence allocation. Nevertheless, the outcomes measured by the investigators were not detailed to the participants. In addition, data recording and data extraction were fully automated. Furthermore, sequence allocation was coded before data was sent to the statistician for analysis.

### 2.2. Outcomes

The primary outcome was the alveolar ventilation during the one-minute cardiopulmonary resuscitation (CPR) scenario according to the sequence used.

To determine alveolar ventilation, the dead space volume was subtracted from each ventilation. According to the appropriate Best Guess formula, a 9-month-old infant should weigh around 9 kg (0.5 × age in months + 4.5) [21]. Using the formula proposed by Numa and Newth [22], this corresponds to a dead space of around 25 mL.

The secondary outcomes were:The total number of ventilations;The proportion of ventilations within (30–70 mL), above (>70 mL), and below (<30 mL) the target volume (according to the manikin’s manufacturer);The alveolar ventilation obtained without taking ventilation volumes over 70 mL into account (for this analysis, all ventilations were individually capped at 45 mL);The proportion of adequate chest compression depth according to the manikin’s size (≥4.3 cm, corresponding to one-third of the height of the manikin’s chest, i.e., 13 cm);The proportion of chest compressions within (100–120 compressions per minute, cpm), above (>120 cpm), and below (<100 cpm) the target rate;The chest compression fraction (CCF);The proportion of compressions with adequate chest recoil.

Because the pre-determined chest compression depth target was not reached during this study, an additional chest compression depth outcome considering an arbitrary target (≥3 cm) based on the mean compression’s depth rather than the guidelines was added post hoc.

### 2.3. Data Extraction, and Availability

Data were automatically collected through the manikin’s sensors, thereby preventing assessment bias. All the variables of interest listed above were automatically generated using a custom-coded hypertext pre-processor (PHP) script. These variables were then exported to a comma-separated values (CSV) file and imported for statistical analysis in Stata 15.1 (StataCorp, LLC, College Station, TX, USA). All authors had access to the database. The curated dataset is publicly available on Yareta [23].

### 2.4. Sample size

Sample size calculation could only be based on an estimate, since little relevant data were available to compute it for such a study. To be conservative, a correlation of 0 was used. Assuming a mean tidal volume of 50 mL, the mean alveolar volume would be of 25 mL. with 10 expected ventilations in the AHA group versus 12 in the ERC group, there would be a difference of 50 mL in alveolar ventilation during the first minute (250 mL of alveolar ventilation per minute in the AHA group versus 300 mL in the ERC group). Variability should be high, and a standard deviation of 50 was therefore used. with an alpha set at 5% and for a power of 90%, the required sample size was 24 participants (48 simulations). Given the study design (simulation) and the setting (participants of the first Swiss Prehospital Research Day), more participants were accepted.

### 2.5. Bias Minimization and Statistical Analysis

This study was designed to avoid all potential sources of carryover effects [24]. One of the most common carryover effects found in cross-over simulation studies is related to the fact that participants sometimes learn how to perform a task during the first simulation sequence. The enrollment process, however, ensured that all participants were proficient in BLS maneuvers, including BVM oxygenation and ventilation. In addition, all participants were given the opportunity to practice these skills on a SimBaby manikin identical to the one used to record resuscitation data, with no time limit, prior to carrying out their allocated resuscitation sequences. Furthermore, the equipment available to practice was identical to that present in the study room. Moreover, any residual learning effect would have been further mitigated by the duration of the resuscitation sequences, each lasting one minute. These short durations should also have prevented the onset of fatigue.

Though unlikely because of the aforementioned precautions, any remaining carryover effects would have been evened out by virtue of complete counterbalancing [24]. Indeed, the randomization mechanism ensured a 1:1 distribution of resuscitation sequences (AHA → ERC or ERC → AHA), which are both of equal magnitude. In light of these elements, no specific statistical analyses were performed to detect a carryover effect, but the dependency of the variables was taken into account through the use of paired tests. Normality was assessed graphically and was considered doubtful. Given the limited sample size, non-parametric tests were therefore used, and the data were described using the median [Q1;Q3]. A two-sided p-value lower than 0.05 was considered significant.

## 3. Results

A total of 28 participants registered, and 56 resuscitation sequences were recorded and analyzed. All participants worked exclusively in the prehospital setting. Their other characteristics are detailed in Table 1.

### 3.1. Primary Outcome

Minute alveolar ventilation was higher when applying the ERC approach (370 mL [203–472] versus 276 mL [140–360], *p* < 0.001) (Figure 2).

### 3.2. Secondary Outcomes

The secondary outcomes are displayed in Table 2.

## 4. Discussion

This simulation study confirms that the ERC guidelines enable higher alveolar ventilation volumes during the first minute of resuscitation maneuvers. This could be considered hardly surprising, since the sheer number of ventilations should be significantly higher when applying these guidelines. Nevertheless, ventilations are not necessarily effective, and their volume can markedly vary from one ventilation attempt to another. A recent simulation study by Santos-Folgar et al. showed that insufflations delivered using BVM devices failed to reach the alveoli 14% of the time and that the volume of 72% of these ventilations was considered insufficient [25]. In the present study, ventilations were applied by professional prehospital providers, and the proportion of ventilations below the 30 mL target were very low (less than 5%). Less-experienced providers, rescuers belonging to non-medical emergency services, non-professional first responders, and lay people may all be less successful at providing adequate ventilations.

The CCF was unsurprisingly lower when ERC guidelines were applied, but the difference, which was less than 10%, was smaller than would be expected given the time required to perform the initial five ventilations. Regardless of the guidelines followed, the CCF was consistently low, with the third quartile value not even reaching 70%, even when AHA guidelines were applied. The association between CCF and return of spontaneous circulation is well established in adults, and CCFs higher than 80% are advisable in case of adult OHCA [26,27]. Since participants were unaware of the outcomes studied, these low CCF values deserve further attention, and means of improving them should actively be sought.

The cut-off used to determine whether chest compression depth was adequate was decided according to the manikin’s size. Since the initial analysis showed that compression depth was consistently shallower than expected, a supplementary secondary outcome was added post hoc using an arbitrary target of ≥3 cm (based on the mean compressions’ depth) to define adequate compression depth. No significant difference was found depending on the guidelines used, but the issue of manikin fidelity should nevertheless be considered. Indeed, high-fidelity simulations have been shown to improve compression quality [28], and manikins’ limitations should be clearly acknowledged by their manufacturers, who should strive to increase the fidelity of their simulation materials.

Although statistically significant, the differences in alveolar ventilation and in CCF during the first minute of resuscitation maneuvers are far from overwhelming and thus may not be clinically meaningful. Furthermore, these differences may well fade after a few minutes of CPR. This raises the question of whether a specific pediatric OHCA algorithm should be maintained and promoted. Indeed, the ABC (airway, breathing, chest compressions) mnemonic was replaced by the CAB one to place more emphasis on chest compressions, since establishing a patent airway and providing effective ventilations was time-consuming and markedly delayed the provision of effective resuscitation maneuvers [29,30]. There is little reason to believe that most rescuers are currently more proficient in airway management than they were a decade ago and that reverting to the old mnemonic would be associated with improved times. In addition, one of the main challenges faced by policy makers and public health authorities is to increase the rate of bystander CPR after OHCA [31]. Bystander CPR rates vary widely from one region to another [32,33]. The theory of planned behavior, which states that the probability of carrying out an action depends on the intention of performing it, is often used to explore the reasons underlying the differences in bystander CPR rates [34,35]. One of the 3 dimensions of this theory, controlled beliefs, relates to confidence in one’s ability to carry out an action, in this case CPR. This confidence could be dampened by the multiplicity of resuscitation guidelines. Therefore, promoting specific guidelines would only be appropriate if they provide indisputable advantages. While the ERC approach somewhat enhanced alveolar ventilation during the first minute of resuscitation maneuvers, it also impaired the CCF, and the differences reported can hardly be considered compelling. Even though this is only a simulation study, all the aforementioned elements should be taken into account by the ERC when deciding whether to continue supporting a pediatric-specific OHCA guideline.

This study has several limitations. First, this was a simulation study, and no actual clinical outcomes could therefore be measured. Second, even though a sample size calculation was performed, and the required number of participants exceeded, the sample size was limited. This limitation is however mitigated by the fact that results were mostly consistent, with little variability between observations. Therefore, there would likely have been little benefit to increasing the number of participants. Another limitation is that these resuscitation sequences were only carried out by professional prehospital providers. It is doubtful that less experienced providers would obtain the same results, and further studies could therefore be considered, since data are sparse regarding differences in ventilation quality according to rescuer background [36,37]. Finally, the resuscitation scenario used in the course of this study did not reflect actual prehospital conditions, and the time needed to initiate ventilation maneuvers was not assessed, since it would not have provided meaningful data. However, it is reasonable to think that the ERC algorithm could significantly delay the first chest compression. 

There are also many strengths to this study, including the randomization process, the blinding of the participants as to the outcomes studied, the automatic data recording and extraction process, and the blinding of the data analyst. Another strength is the use of a little studied yet clinically relevant physiological outcome, i.e., alveolar ventilation, while also measuring all standard resuscitation parameters. 

## 5. Conclusions

In this pediatric OHCA simulation study, the ERC approach enabled higher alveolar ventilation volumes at the cost of lower chest compression fractions. The opposite was true with the AHA approach. Although statistically significant, the difference in alveolar ventilation may not be clinically relevant. Since neither approach can be considered unquestionably superior to the other, and because of the importance of overcoming barriers to resuscitation, promoting different resuscitation algorithms for children and for adults may not be appropriate.

## Figures and Tables

**Figure 1 healthcare-10-02451-f001:**
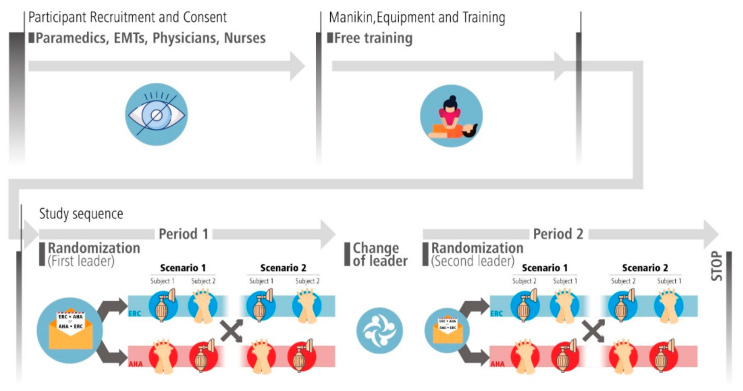
Study sequence. Teams of two participants will first train on a manikin identical to the one used to collect data. After entering the study room, the first leader will pick an opaque, sealed envelope, and carry out two resuscitation sequences according to the allocation. Team members will then switch roles. The new leader will then pick another envelope and perform the two last resuscitation sequences accordingly.

**Figure 2 healthcare-10-02451-f002:**
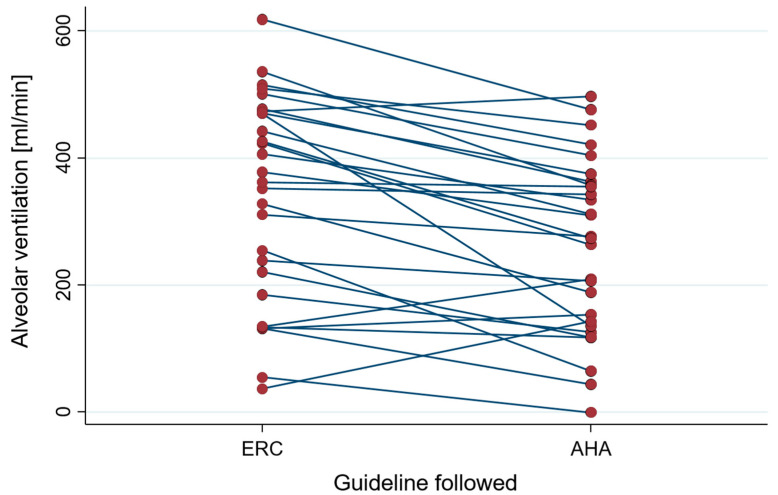
Alveolar ventilation during the first minute of resuscitation.

**Table 1 healthcare-10-02451-t001:** Characteristics of study participants.

Characteristic	Participants (N = 28)
Gender (*n*, %)	
*Man*	19 (68%)
*Woman*	9 (32%)
Age (years, median, [Q1;Q3])	32 [27;45]
Profession (*n*, %)	
*Paramedic*	27 (96%)
*Physician*	1 (4%)
Years since graduation (median, [Q1;Q3])	7 [2;17]
Years of prehospital experience (median, [Q1;Q3]) ^1^	13 [2;18]
Time elapsed since last pediatric resuscitation in the field (*n*, %) ^2^	
*Less than 6 months ago*	0 (0%)
*6–12 months ago*	3 (11%)
*12–24 months ago*	6 (21%)
*More than 24 months ago*	7 (25%)
*No prior pediatric resuscitation in the field*	12 (43%)
Time elapsed since last simulated pediatric resuscitation (*n*, %) ^2^	
*Less than 6 months ago*	13 (46%)
*6–12 months ago*	8 (29%)
*12–24 months ago*	3 (11%)
*More than 24 months ago*	4 (14%)
*No prior simulated pediatric resuscitation*	0 (0%)
Specific post-graduate pediatric course followed (yes, *n*, %)	17 (61%)

^1^ in Switzerland, most paramedics work in the prehospital setting before graduating (either in the course of their studies or as less-qualified providers); ^2^ Totals may not equal 100% due to rounding.

**Table 2 healthcare-10-02451-t002:** Secondary outcomes, expressed as median [Q1;Q3].

Outcome	ERC Approach	AHA Approach	Difference
Number of ventilations	13 [12;15]	10 [8;10]	3.5 [3;5]
Ventilation’s volume	54 mL [37;61]	52 mL [43;63]	−1 mL [−6;3]
Proportions of ventilations			
- *Below target (<30 mL)*	4% [0;23]	0% [0;11]	0% [−2;7]
- *In target (30–70 mL)*	76% [65;82]	75% [52;100]	2% [−9;10]
- *Above target (>70 mL)*	3% [0;24]	0% [0;31]	0% [−3;0]
Alveolar ventilation with ventilation capped at 70 mL	365 mL [203;445]	271 mL [138;353]	78 mL [33;117]
Compressions’ depth	32 mm [28;34]	32 mm [30;35]	−1 mm [−2;1]
Proportions of compressions with adequate depth			
- *According to the manikin’s size (≥4.3 cm)*	0%	0%	0%
- *According to the ≥3 cm target*	91% [17;100]	89% [36;99]	0% [−15;3]
Compression rate	109 cpm [103;114]	110 cpm [104;114]	−1 cpm [−3;1]
Proportions of compression rate			
- *Below target (<100 cpm)*	0% [0;20]	3% [0;13]	0% [−2;2]
- *In-target (100–120 cpm)*	91% [57;98]	90% [55;96]	1% [−5;6]
- *Above target (>120 cpm)*	1% [0;7]	0% [0;9]	0% [−1;1]
CCF	57% [54;64]	66% [59;68]	−7% [−11;−2]
Proportion of compressions with adequate chest recoil	93% [42;100]	76% [34;92]	6% [−8;20]

AHA: American Heart Association; CCF: chest compression fraction; cpm: compressions per minute; ERC: European Resuscitation Council.

## Data Availability

Study data is publicly available on Yareta [23].

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
