# Peer review of "Impact of Two Resuscitation Sequences on Alveolar Ventilation during the First Minute of Simulated Pediatric Cardiac Arrest: Randomized Cross-Over Trial"

_healthcare, 2022, doi:10.3390/healthcare10122451_

Round 1

Reviewer 1 Report (New Reviewer)

It is very interesting and unique. 

I agreed that difference of AHA and ERC spawns from a lack of definitive evidence which prevented the ILCOR from issuing an unequivocal recommendation.

Your attempt was good. But I suggest that you would change the conclusion a little. The conclusion should include your results.

Author Response

Dear Reviewer,

Thank you very much for your review. Here are the responses to your comments:

Comment 1: It is very interesting and unique.
Response: Thank you for this heart warming comment.

Comment 2: I agreed that difference of AHA and ERC spawns from a lack of definitive evidence which prevented the ILCOR from issuing an unequivocal recommendation.
Response: We could not agree more, and hope that our results will help the ILCOR issue an unequivocal recommendation.

Comment 3: Your attempt was good. But I suggest that you would change the conclusion a little. The conclusion should include your results.
Response: Thank you for your suggestion. The conclusions have been modified thus: "In this pediatric OHCA simulation study, the ERC approach enabled higher alveolar ventilation volumes at the cost of lower chest compression fractions. The opposite was true with the AHA approach. Although statistically significant, the difference in alveolar ventilation may not be clinically relevant. Since neither approach can be considered unquestionably superior to the other, and because of the importance of overcoming barriers to resuscitation, promoting different resuscitation algorithms for children and for adults may not be appropriate."

Reviewer 2 Report (New Reviewer)

Thank you for the opportunity to review this manuscript. This study is well-designed and conducted. However, I have several points for the authors to consider.

First, please describe why the authors focused on the “alveolar ventilation” of infant resuscitation. With more references on alveolar ventilation associated with improving outcome, readers can better understand the study results and may consider taking this study results into their own practices.

Second, similarly, the discussion part should focus more on the results of alveolar ventilation if the authors proceed with the study purpose. It seems the authors focused too much on other outcome variables such as compression depth and CCF. These variables should not difference between the algorithms. 

This study showed that the impact of alveolar ventilation with ERC pediatric resuscitation procedure was minor when compared to AHA procedure. The authors concluded that promoting these two different algorithms may not be appropriate.

Along with the study results, both algorithms provide same quality procedures on ventilation and chest compressions. It seems okay to say that there is no big difference between the algorithms in terms of resuscitation procedures. However, ERC algorithm may significantly delay the first chest compression. With this time variable, the authors can provide significant information to readers.

Author Response

Dear Reviewer,

Thank you very much for this review. Here are our responses:

Comment 1: Thank you for the opportunity to review this manuscript. This study is well-designed and conducted. However, I have several points for the authors to consider.
Response: Thank you for this general comment. We were delighted to read your comments.

Comment 2: First, please describe why the authors focused on the “alveolar ventilation” of infant resuscitation. With more references on alveolar ventilation associated with improving outcome, readers can better understand the study results and may consider taking this study results into their own practices.
Response: Thank you for this comment. In the discussion, we acknowledged that alveolar ventilation is "a little studied yet clinically relevant physiological outcome". The physiological aspects are detailed in the introduction: "Alveolar ventilation is the main determinant of CO2 clearance [10], which is necessary to improve alveolar oxygen partial pressure [11]. This partial pressure is finally responsible for adequate tissue oxygenation [...]". We have modified the third paragraph of the introduction to acknowledge your comment. It now reads: "The impact of alveolar ventilation on newborns has previously been described [13,14], but there is little data regarding the impact of resuscitation sequences on alveolar ventilation in somewhat older infants." 2 references have been added:
- Milner, A. The importance of ventilation to effective resuscitation in the term and preterm infant. Semin. Neonatol. 2001, 6, 219–224, doi:10.1053/siny.2001.0057.
- Miller, J.D.; Carlo, W.A. Safety and effectiveness of permissive hypercapnia in the preterm infant. Curr. Opin. Pediatr. 2007, 19, 142–144, doi:10.1097/MOP.0b013e3280895e12.

Comment 3: Second, similarly, the discussion part should focus more on the results of alveolar ventilation if the authors proceed with the study purpose. It seems the authors focused too much on other outcome variables such as compression depth and CCF. These variables should not difference between the algorithms. 
Response: Thank you for this comment. The discussion starts by stating that "This simulation study confirms that the ERC guidelines enable higher alveolar ventilation volumes during the first minute of resuscitation maneuvers.". In line with your comment, we have added a reference to a recently published article (Santos-Folgar, M.; Lafuente-Filgueira, P.; Otero-Agra, M.; Fernández-Méndez, F.; Barcala-Furelos, R.; Trastoy-Quintela, J.; Aranda-García, S.; Fernández-Méndez, M.; Rodríguez-Núñez, A. Quality of Ventilations during Infant Resuscitation: A Simulation Study Comparing Endotracheal Tube with Face Mask. Children 2022). The following sentence was added: "A recent simulation study by Santos-Folgar et al. showed that insufflations delivered using BVM devices failed to reach the alveoli 14% of the time and that the volume of 72% of these ventilations was considered insufficient [25]."

Comment 4: This study showed that the impact of alveolar ventilation with ERC pediatric resuscitation procedure was minor when compared to AHA procedure. The authors concluded that promoting these two different algorithms may not be appropriate. Along with the study results, both algorithms provide same quality procedures on ventilation and chest compressions. It seems okay to say that there is no big difference between the algorithms in terms of resuscitation procedures. However, ERC algorithm may significantly delay the first chest compression. With this time variable, the authors can provide significant information to readers.
Response: You are of course perfectly right. However, this variable was not measured in the course of this study, and we have modified the discussion to make this fact (and the reasons underlying the decision not to measure this variable) clearer: "Finally, the resuscitation scenario used in the course of this study did not reflect actual prehospital conditions, and the time needed to initiate ventilation maneuvers was not assessed since it would not have provided meaningful data. However, it is reasonable to think that the ERC algorithm could significantly delay the first chest compression."

Reviewer 3 Report (New Reviewer)

The study is regarding with the randomized comparison of ERC and AHA pediatric CPR guidelines. The paper is well written but there are serious gaps in the methodology.

It was stated that "Each team of two performed a total of 4 resuscitation sequences, each lasting one minute." It is obvious that if a CPR attempt includes more rescue breath in one minute, it has more alveolar ventilation and less chest compression. However, if it lasts 10 minutes, these differences cannot be meaningful. On the other hand, in clinical situations, no CPR attempt lasts one minute. Maybe, if the length of resuscitation would be more, we would understand more clearly when these differences become meaningless. This methodological approach seriously impacts the study's contribution to the literature and will also lead to uncertainty about how the results will affect clinical practice.

For the conclusion of the study, it wouldn’t be purely correct with the results of one minute CPR to suggest the same algorithm for guidelines in addition to the comments above.

In the study, almost entirely paramedics completed CPR scenario. The results are unknown what if other healthcare professionals would performed the CPR.

It is not clearly understood how total ventilation for each rescue breath was calculated in the study? From manikin data set.

I am not sure to use “cardiac arrest” term for pediatric cases if it commonly results from pulmonary reasons.

Author Response

Dear Reviewer,

Thank you for the time and effort you spent reviewing our manuscript. Here are the responses to your comments:

Comment 1: The study is regarding with the randomized comparison of ERC and AHA pediatric CPR guidelines. The paper is well written but there are serious gaps in the methodology. It was stated that "Each team of two performed a total of 4 resuscitation sequences, each lasting one minute." It is obvious that if a CPR attempt includes more rescue breath in one minute, it has more alveolar ventilation and less chest compression. However, if it lasts 10 minutes, these differences cannot be meaningful. On the other hand, in clinical situations, no CPR attempt lasts one minute. Maybe, if the length of resuscitation would be more, we would understand more clearly when these differences become meaningless. This methodological approach seriously impacts the study's contribution to the literature and will also lead to uncertainty about how the results will affect clinical practice.
Response: Thank you for this thought-provoking comment. While we of course agree with you that more rescue breaths would be performed using the ERC sequence (this was used to perform the sample size calculation, see sectinon 2.4 - Sample Size), the 5  initial ventilations provided before initiating chest compressions might have been more effective or less effective than those provided under the 15:2 regimen. The issue of ventilation volume was recently assessed by Santos-Folgar et al. (Santos-Folgar, M.; Lafuente-Filgueira, P.; Otero-Agra, M.; Fernández-Méndez, F.; Barcala-Furelos, R.; Trastoy-Quintela, J.; Aranda-García, S.; Fernández-Méndez, M.; Rodríguez-Núñez, A. Quality of Ventilations during Infant Resuscitation: A Simulation Study Comparing Endotracheal Tube with Face Mask. Children 2022). The following sentence was added: "A recent simulation study by Santos-Folgar et al. showed that insufflations delivered using BVM devices failed to reach the alveoli 14% of the time and that the volume of 72% of these ventilations was considered insufficient [25]." after the sentence "Nevertheless, ventilations are not necessarily effective and their volume can markedly vary from one ventilation attempt to another.".
Of course, with time, any difference should and probably would be evened out with time. This is ackowledged thus: "Furthermore, these differences may well fade after a few minutes of CPR.".

Comment 2: For the conclusion of the study, it wouldn’t be purely correct with the results of one minute CPR to suggest the same algorithm for guidelines in addition to the comments above.
Response: We acknowledge that our results are all but incontrovertible and have therefore carefully phrased our conclusion. We nevertheless emphasized the "importance of overcoming barriers to resuscitation", and believe that a unique algorithme may somewhat help in this regard.

Comment 3: In the study, almost entirely paramedics completed CPR scenario. The results are unknown what if other healthcare professionals would performed the CPR.
Response: You are of course right. This is acknowledged in the limitations section of the discussion: "Another limitation is that these resuscitation sequences were only carried out by professional prehospital providers. It is doubtful that less experienced providers would obtain the same results, and further studies could therefore be considered since there is little data regarding differences in ventilation quality according to rescuer background [36,37]."

Comment 4: It is not clearly understood how total ventilation for each rescue breath was calculated in the study? From manikin data set.
Response: This was addressed under section 2.3 (Data Extraction, and Availability): "Data was automatically collected through the manikin’s sensors, thereby preventing assessment bias" and under section 2.2 (Outcomes): "To determine alveolar ventilation, the dead space volume was subtracted from each ventilation. According to the appropriate Best Guess formula, a 9-month old infant should weigh around 9 kg (0.5 x age in months + 4.5) [21]. Using the formula pro-posed by Numa and Newth [22], this corresponds to a dead space of around 25 ml.". We refrained from adding the PHP code to the manuscript since we believe that it would not be relevant to most readers, but you can access the original PHP script here: https://data.mendeley.com/datasets/s8d2gpfhyw/1 (this is the script used in Stuby et al,  Effect of Early Supraglottic Airway Device Insertion on Chest Compression Fraction during Simulated Out-of-Hospital Cardiac Arrest: Randomised Controlled Trial. J. Clin. Med. 2022 - https://doi.org/10.3390/jcm11010217)

Comment 5: I am not sure to use “cardiac arrest” term for pediatric cases if it commonly results from pulmonary reasons.
Response: This is, without a doubt, a good question. After reviewing the ILCOR guidelines, it does indeed seem that "cardiac arrest" should be used in this context (see https://costr.ilcor.org/document/?category=pediatrics).

Reviewer 4 Report (New Reviewer)

11.   General comments:

I believe that the problem of the article is relevant and current in the topic of Pediatric Cardiac Arrest and life support.

  1. Title

The title reflects the content and problem studied and the type of the study

  1. Abstract

The abstract complies with the Journal's standards. It is based on international organizations (ILCOR, AHA).

It refers to the study population and emphasizes the (adequate) methodology and the main results obtained.

The conclusions are clear and well synthesized

4.     Keywords

The keywords are representative of the subject studied and exposed

5.     Introduction

A state of the art is carried out in relation to the study. The objective of the study is mentioned, as well as the justification for the choice and importance of studying this topic. Mention the main international organizations ILCOR...

6.     Materials and Methods

The work methodology is precise. The design is appropriate to the objectives pursued, mentions the study population. The pediatric population is well defined. The work is replicable. The drawings are explanatory, they are suitable in this section. Double blind. Very good methodology

7.     Results

The results shown are concrete and detailed, explaining how to obtain this information and what scientific evidence it has. Although it does not present a very large sample, it is sufficient for this study given its methodology and novelty. The results are clearly defined, showing the different sociodemographic parameters as well as those derived from life support (ventilatory volume, proportion of ventilations...etc.). The results are presented appropriately

  1. Discussion

The discussion is adequate, the authors compare their results with those obtained in other similar studies. The implications of the study are reasoned as well as its importance.

9.     Conclusions

The conclusions show the main findings found. It is presented correctly

  1. References

The references are adequate and up-to-date, most of them from the last 3 years. See the main publications on this topic

Author Response

Dear Reviewer,

We were delighted to read your comments and thank you very much for your thorough review.

Round 2

Reviewer 2 Report (New Reviewer)

Thank you for revising the manuscript.

This manuscript is a resubmission of an earlier submission. The following is a list of the peer review reports and author responses from that submission.

Round 1

Reviewer 1 Report

It is a clearly presented study design that would add immense value to the journal

Author Response

Dear Reviewer,

Thank you very much for your extremely positive comment.

Reviewer 2 Report

The idea of the paper is very good although the method has some limitations.

My main concern regarding this paper keeps on the rationale of publishing a such protocol. I can understand the rationale of a publishing a protocol before the start of a study but in the case of studies that enrolls many patients and it lasts for a long time (months, years). There also are many sites dedicated for publishing such protocols (e.g. https://clinicaltrials.gov/). 

In this case we talk about a study performed in one day (01.09.2022) that includes 24 participants.

I suggest to authors to improve the protocol with more scenarios of cardiac arrest following bronchiolitis, arrhythmias, congenital heart disease. I consider that oxygenation and CO2 removal could have different impact on the outcome depending the type of the causes, this should be check in this study.

The authors mentioned "This protocol is version 0.5 (18.07.2022). The version submitted for publication will be 1.0." When will we see the version 1.0?? The authors intend to submit another paper with the final version of the protocol?

The authors also affirmed "This will be a randomized, cross-over, superiority trial.." Why this trial is superior? Or the sentence is not finished?

Author Response

Dear Reviewer,

Thank you for your timely and thought-provoking review.

We agree that questioning the rationale of publishing a study protocol is of paramount importance, and all Reviewers should systematically consider whether a particular paper is worth publication.

In the case of this simulation study, the protocol has been registered on clinicaltrials.gov and there are of course no patients involved. You are therefore perfectly right regarding the fact that publishing this protocol will hardly affect the risk of reporting bias and that it will not result in a decreased probability of harm to the participants.

However, publishing study protocols can also serve other purposes (Cheng et al, Designing and Conducting Simulation-Based Research - doi: 10.1542/peds.2013-3267). In our specific case, we believe that this manuscript is one of the few scientific papers acknowledging the difference between AHA and ERC guidelines regarding the initial pediatric OHCA resuscitation sequence. In addition, the primary outcome which will be used and reported (i.e., mean minute alveolar ventilation) has seldom been considered, even though it could be more physiologically relevant than other, more commonly reported ventilation outcomes. We strongly believe that reporting these elements could be of significant interest to fellow clinicians and researchers. While other manuscript formats (viewpoint, lettrer, etc.) could have been considered to convey our thoughts, we think that outlining these elements and determining a method to gather relevant (but of course not definitive) evidence allowed us to go one step further. Moreover, we are convinced that submitting a protocol for peer review is the best way of obtaining constructive feedback from fellow researchers, and such feedback frequently leads to the improvement of such protocols. By submitting our manuscript to an MDPI journal (in this case Medicina), we were able to gather 3 independent, high-quality reports in a timely manner.

Finally, we think that the decision of whether to publish study protocols rests with the editorial board of each journal. There is a progressive trend towards protocol publication, and many authors have contributed manuscripts advocating the publication of such manuscripts (Ohtake PJ & Childs JD, Why Publish Study Protocols? - doi: 10.2522/ptj.2014.94.9.1208; Kim SY, Why do journals publish research
protocols? - 10.6087/kcse.280; ...).  We will therefore await the Editorial Decision as to the publication of our manuscript and thank you once again for the time and effort you spent reviewing our manuscript.

Here are the responses to the specific points you raised:

Comment 1: My main concern regarding this paper keeps on the rationale of publishing a such protocol. I can understand the rationale of a publishing a protocol before the start of a study but in the case of studies that enrolls many patients and it lasts for a long time (months, years). There also are many sites dedicated for publishing such protocols (e.g. https://clinicaltrials.gov/). 
Response: We hope that our introduction has helped explain our position. We considered adding a paragraph to the discussion to acknowledge our stance, but finally decided not to since this would go beyond the scope of this particular protocol. Nevertheless, we have chosen the "open peer review" option proposed by MDPI journals, and both your comments and our responses will be publicly available provided our manuscript is finally accepted.

Comment 2: I suggest to authors to improve the protocol with more scenarios of cardiac arrest following bronchiolitis, arrhythmias, congenital heart disease. I consider that oxygenation and CO2 removal could have different impact on the outcome depending the type of the causes, this should be check in this study.
Response: There was considerable debate among the authors regarding the scenario the partipants would be confronted with. Consistently with our objective (i.e., "to determine the difference in alveolar ventilation during the first minute of resuscitation according to the sequence used (AHA vs ERC) in a pediatric model of OHCA"), we decided to stick to a simple scenario (section 2.1.4) since multiplying scenarios could lead to inconsistent behavior among participants and would introduce a significant bias. Depending on our results, further studies could then assess the impact of the initial minute alveolar ventilation in specific pathologies. This should be discussed in the results paper.

Comment 3: The authors mentioned "This protocol is version 0.5 (18.07.2022). The version submitted for publication will be 1.0." When will we see the version 1.0?? The authors intend to submit another paper with the final version of the protocol?
Response: We apologize for this oversight and have now removed this sentence, which has been replaced by "This trial has been registered on clinicaltrials.gov (NCT05474170)." We have no intention of submitting another paper with another version of this protocol.

Comment 4: The authors also affirmed "This will be a randomized, cross-over, superiority trial.." Why this trial is superior? Or the sentence is not finished?
Response: We believe that this sentence is self-standing since this will indeed be a superiority trial (10.1016/j.prrv.2019.06.002). This reference has now beeen added to the manuscript to clarify this point.

Reviewer 3 Report

Dear authors, thank you very much for your manuscript, which I read with interest.

Your protocol is well written in each of its paragraphs. The introduction nicely presents the differences between the two guidelines for resuscitation in case of paediatric cardiac arrest and clearly states the aim of your study by describing thoroughly and with appropriate citations the reasons which move you to perform this simulation study.

The protocol presents a detailed chapter of materials and methods, which allows the readers to understand your scientific approach and that will be of support for other researchers who may perform other simulation studies. I really appreciated the criticism you move towards yourselves presenting the limitations of the methodology in the discussion. In my opinion, the protocol is interesting and, considering the study design, has a correct scientific approach.

While reading the manuscript, I found a couple of mistakes in text editing, that require a minor revision, please provide corrections:

Paragraph 2.1: In the first line there are two fullstops. (..)

Paragraph 2.1.3: Bag Valve Mask is reported directly as an acronym, please add the complete name and report BVM in brackets.

Author Response

Dear Reviewer,

Thank you very much for the time and energy you put into this review. Here are the responses to your comments:

Comment 1: Your protocol is well written in each of its paragraphs. The introduction nicely presents the differences between the two guidelines for resuscitation in case of paediatric cardiac arrest and clearly states the aim of your study by describing thoroughly and with appropriate citations the reasons which move you to perform this simulation study.
Response: Thank you very much.

Comment 2: The protocol presents a detailed chapter of materials and methods, which allows the readers to understand your scientific approach and that will be of support for other researchers who may perform other simulation studies. I really appreciated the criticism you move towards yourselves presenting the limitations of the methodology in the discussion. In my opinion, the protocol is interesting and, considering the study design, has a correct scientific approach.
Response: Thank you once again for your very constructive comments. We strongly believe that outlining potential (and actual) limitations helps researchers circumvent those which could be avoidable, and consider alternative or supplementary designs to address such limitations.

Comment 3: Paragraph 2.1: In the first line there are two fullstops. (..)
Response: Thank you for spotting this mistake, which has now been addressed.

Comment 4: Paragraph 2.1.3: Bag Valve Mask is reported directly as an acronym, please add the complete name and report BVM in brackets.
Response: You are perfectly right, we apologize for this oversight and have now added the complete meaning of this acronym.

Round 2

Reviewer 2 Report

Dear authors,

I appreciate your elegant response. Although some corrections have been done to the first version of the manuscript i have still major concerns. I have not a problem of publishing a protocol as i have said (although your reference Cheng et al. is a review about simulation not a protocol study!).

Your idea is good, to compare the protocols using simulators, but your approach is too simple to be published in a journal with an IF = 2.9!

So i ask you again to improve your clinical scenarios of cardiac arrest (i have already suggested you in previous report). I consider that is not a difficult task for your team.

Otherwise i consider that the manuscript it will be not suitable for publishing in this Journal (maybe other journal with a lower IF would accept it).

Regards, 

Author Response

Dear Reviewer,

Thank you for re-reviewing our manuscript. Here is the response to your major concern:

Comment: So i ask you again to improve your clinical scenarios of cardiac arrest (i have already suggested you in previous report). I consider that is not a difficult task for your team.
Response: We have heeded your advice and section 2.1.4 - Resuscitation scenario - has been markedly extended. It now reads: "Participants will be told that they are facing a 9-month old infant who suddenly collapsed. To be consistent with the most common causes of cardiac arrest among infants, i.e., hypoxia, the monitor will display a heart rate of 65 per minute throughout the scenario [16]. Therefore, participants will face a pulseless electrical activity scenario and will be able to ask questions to an investigator who will act as the child’s parent. This investigator will tell the participants that foreign body airway obstruction is highly unlikely and that the infant did not display any sign compatible with such an obtruction prior to collapsing.  The duration of this scenario will prevent epinephrine administration, but participants will nevertheless be able to ask questions to explore the cause of the cardiac arrest [17]. If asked, the parent will tell the participants that the infant was born at full term with a normal birth weight, and that there were no complications during vaginal delivery. Except for a cough which was present 2 days prior to the cardiac arrest, the infant was in good health, with no known allergies, and was not given any medication. If prompted, the parent will inform the participants that the infant is breastfed, with the last feeding attempt having taken place 3 hours before the event."

The following references were also added:

  • Luong D, Cheung PY, Barrington KJ, Davis PG, Unrau J, Dakshinamurti S, Schmölzer GM. Cardiac arrest with pulseless electrical activity rhythm in newborn infants: a case series. Arch Dis Child Fetal Neonatal Ed. 2019 Nov;104(6):F572-F574. doi: 10.1136/archdischild-2018-316087. Epub 2019 Feb 22. PMID: 30796058.
  • Vega RM, Kaur H, Edemekong PF. Cardiopulmonary Arrest In Children. 2022 May 22. In: StatPearls [Internet]. Treasure Island (FL): StatPearls Publishing; 2022 Jan–. PMID: 28613789.